# The Effect of Prior Mental Health on Persistent Physical Symptoms after Exposure to a Chemical Disaster

**DOI:** 10.3390/healthcare11071004

**Published:** 2023-03-31

**Authors:** Young-Sun Min, Soo-Young Kim, Sun-Kyeong Choi, Yeon-Soon Ahn

**Affiliations:** 1Department of Occupational and Environmental Medicine, Soonchunhyang University Cheonan Hospital, Cheonan 31151, Republic of Korea; mys0303@schmc.ac.kr; 2Affiliated Clinic, LG Energy Solution, Cheongju 28122, Republic of Korea; 3Department of Preventive Medicine and Genomic Cohort Institute, Yonsei University Wonju College of Medicine, Wonju 26426, Republic of Korea; seonky@yonsei.ac.kr

**Keywords:** chemical accidents, chemical hazard release, health surveys, mental health, medically unexplained symptoms, styrene monomer

## Abstract

A styrene monomer (SM) oil vapor leak occurred at a chemical plant in Seosan, South Korea on 17 May 2019. A bad odor developed, and many residents complained of various symptoms and visited nearby medical institutions. We analyzed the demographic and clinical characteristics of patients treated at local hospitals and clinics for symptoms related to SM exposure, and identified factors affecting symptom persistence in any organ. Data were collected by the main Seosan office, and 1201 (33.0%) subjects agreed to participate in this study. We used the Assessment of Chemical Exposure toolkit of the Agency for Toxic Substances and Disease Registry. Logistic regression was performed to determine whether mental health symptoms prior to the accident were risk factors for symptom persistence. The strongest risk factor for persistence of at least one symptom in any organ was a preexisting mental health symptom (odds ratio [OR] = 5.47, 95% confidence interval [CI]: 2.57–11.65). Persistent symptoms of the nervous (OR = 1.54), musculoskeletal (OR = 1.92), and gastrointestinal (OR = 1.45) systems were observed. Prior mental health symptoms are risk factors for persistent physical symptoms after a chemical disaster. After a disaster, management of individuals with preaccident mental symptoms or disease is needed.

## 1. Introduction

The National Institute of Chemical Safety reported that 677 chemical accidents occurred from 2014 to 2021 in South Korea: 31 workers died and 606 were injured [1]. Furthermore, 10 local residents died and 206 were injured. Thus, about 25% of all casualties affected residents. However, most residents recovered without any problems other than acute skin irritation. In South Korea, a health impact investigation (HII) is conducted when even one resident is admitted to hospital because of events including a chemical accident [2].

A styrene monomer (SM) oil vapor leak occurred at a chemical plant in Seosan (city) at 11:45 a.m. on 17 May 2019, and a second leak occurred at 03:40 a.m. on 18 May 2019. About 74.7 tons of SM leaked [1]. The government investigation team found that the company failed to follow due safety procedures in transfer of SM, while skilled employees in charge of handling SM walked out and were replaced by inexperienced substitute workers at the time of the leak. The maximum diffusion range indicated by the Areal Locations of Hazardous Atmospheres (ALOHA) model [3,4,5] was about 2.8 km (Figure 1). Two hours after the first leak, the monomer air concentration in the plant was 36 ppm, and the level became undetectable after 5 h. No SM was detected in a nearby village 5 h after the first exposure. The United States Environmental Protection Agency (EPA) Acute Exposure Guideline Level (AEGL) was level 1: “Notable discomfort, irritation, or certain asymptomatic non-sensory effects” [6,7,8,9,10]. Acute inhalation of >50 ppm of styrene (AEGL-2) may irritate the nasal mucosa and eyes; a level > 100 ppm may cause skin irritation and central nervous system symptoms [11]. Given the SM atmospheric concentrations at Seosan, the only expected symptom was irritation [6]. As this was a short-term, low-concentration exposure, no resident was poisoned or died. However, a bad odor developed, and many residents complained of various symptoms and over 3600 visited nearby medical institutions.

Literature data on similar domestic or foreign SM accidents are lacking. This HII study was conducted as part of an epidemiological investigation mandated by the Enforcement Decree of the Chemical Substance Act. We assessed the health effects and symptoms of SM exposure and factors affecting symptom persistence. Mental health is known to affect vulnerable populations after disasters. A health impact survey conducted after the Horse River wildfire in Alberta, Canada reported that adolescents who had experienced previous trauma and mothers who were pregnant or postpartum had higher rates of mental illness following the disaster [12,13,14,15]. Therefore, we assessed whether residents with previous poor mental health were more adversely affected when exposed to a chemical disaster.

## 2. Materials and Methods

Up to 25 July 2019, over 3600 residents of Daesan in Seosan visited medical institutions because of SM exposure. Data were collected by the main Seosan office. In October 2019, a temporary health survey center was established in Daesan, and the research team contacted residents who visited the hospital and asked them to visit the survey center. A total of 1201 subjects (33.0%) agreed to participate in this study. Daesan had a population of 14,378 in 2019, so the participation rate was thus 8.4%. The survey ran from October to December 2019, 6 months after the accident.

The Assessment of Chemical Exposure (ACE) toolkit of the Agency for Toxic Substances and Disease Registry (ATSDR) was used in this study [16]. The kit has five modules: exposure location and situation, health status, fire/explosion status, medical treatment, and occupational history. The first module explores the respiration status of the subject at the time of the leak, types of confined space, the strength of any odor, any efforts to move away from the area, and decontamination status. The second module explores the status of the nervous system (7 items), musculoskeletal system (6 items), respiratory system (7 items), cardiovascular system (4 items), gastrointestinal system (9 items), skin (7 items), mental health (6 items), urinary system (2 items), eyes (3 items), nose (2 items), and ears (1 item; 54 items in total). Symptoms present before the accident, any worsening thereof, and symptom persistence for 24 h after the accident and to the time of the investigation were recorded. We assessed whether anxiety, excitement, irritability, fatigue, boredom, insomnia, depression, or hallucinations were present before the accident, and if any such symptoms worsened thereafter. Participants who answered yes to both questions were considered to have had mental health problems before the accident. All subjects self-reported their occupations, locations during exposure, and whether or not to evacuate from the exposed location (evacuation status). Informed consent was obtained from all subjects.

Subjects were grouped according to whether they were occupationally exposed to chemicals. Subject location at the time of the leak was classified as <3, 3–5, 5–9, or ≥10 km (Figure 2). The odor intensity was classified as low, mild, medium, or severe, and evacuation status after smelling the odor was noted. The EPA stated that a monomer concentration of 20 ppm corresponded to AEGL-1 status [6,17].

We recorded symptoms by organ within 24 h of exposure and at the time of the survey, as well as subject characteristics (gender, age, preexisting symptoms or disease) and exposure characteristics (distance from the accident site, occupational exposure, evacuation status). We aimed to identify factors affecting symptoms at the time of the HII survey. All data were analyzed using SPSS for Windows software (ver. 25.0; IBM Corp., Armonk, NY, USA). Age, gender, location at the time of the accident, odor intensity, distance from the accident site, and evacuation status were categorized. Logistic regression analysis was performed to determine whether prior physical and mental symptoms were risk factors for symptom development within 24 h of the accident and persistent symptoms. We adjusted for possible confounding (performance and exposure) factors (location at the time of the accident, odor intensity, distance from the accident site, and evacuation status). We constructed two logistic models: one included all organ symptoms before the accident, while the other did not. A significance level (α) < 0.05 was considered to indicate statistical significance.

## 3. Results

There were 517 males (43.0%) and 632 females (52.6%) in this study. The gender of 52 cases was not recorded (4.3%). The largest age-group was 60–69 years (*n* = 321; 26.7%), and only three subjects were aged < 10 years. The average distance from the accident site was 6.7 km, with 582 (48.5%) subjects between 5 km and 10 km from the leakage accident site and 241 (20.1%) at least 10 km away. When asked about the presence of an odor, 23 (1.9%) subjects responded that they did not smell an odor (low), 47 (3.9%) reported a mild odor, 141 (11.7%) reported a moderate odor, and 990 (82.4%) reported a severe odor (Table 1).

Of the participants who reported at least one symptom within 24 h of the accident, the nervous system was affected in 1141 (95%), the gastrointestinal system in 1099 (91.5%), the respiratory system in 1012 (84.3%), and the eyes in 984 (81.9%). Nausea was the most common symptom (1036; 86.3%) followed by headache (1032; 85.9%) and dizziness (1012; 84.3%). The nervous system was the most commonly affected system among the 102 participants (8.5%) who reported at least one symptom before the accident. Of the participants, 355 (29.6%) reported at least one symptom before the accident. Six months later, nervous system (*n* = 580; 48.3%), mental health (*n* = 508; 42.3%), and eye (*n* = 476; 39.6%) symptoms were the most common (Table 2).

Table 3 presents the average number of symptoms by organ according to mental health symptoms prior to accident. Of the symptoms that occurred within 24 h, the average number of symptoms was 18 among cases without mental health symptoms before the accident, and 23.46 in cases where such symptoms were present. The mean number of symptoms was 5.39 in those without mental health symptoms at the time of the survey, and 10.63 in the group with mental symptoms. The number of symptoms was highest in those with mental health-related symptoms before the accident.

In the logistic regression analysis, a prior symptom in any organ was the strongest risk factor for symptom persistence at the time of the survey. The logistic model adopted two types: including and excluding past physical symptoms by organs before the accident, because the odds ratios of preaccident symptoms were too great to offset other effects. The strongest risk factor for the persistence of at least one symptom in any organ was any prior mental health symptom (odds ratio [OR] = 5.47, 95% confidence interval [CI]: 2.57–11.65). The presence of mental health symptoms was a risk factor for persistence of symptoms of the nervous system (OR = 1.54), musculoskeletal system (OR = 1.92), and gastrointestinal system (OR = 1.45). In the model that excluded past physical symptoms, mental health symptoms were risk factors for symptom persistence in all organs, except the ears, and the overall risk of symptom persistence was high (Table 4).

## 4. Discussion

Disasters include natural disasters, large oil and chemical spills, and traffic pileups. Usually, natural disasters end of their own accord, but the long-term effects of chemical spills are unclear [18]. Although many industrial and environmental disasters involving neurotoxins or radioactive contamination have occurred over the last few decades, the effects on psychological function have been little studied [19]. Very few reports have described the acute effects of exposure to SM. Acute inhalation of >50 ppm SM (AEGL-2) irritates the nasal mucosa and eyes. In amounts > 100 ppm, skin irritation and central nervous system problems develop, with the latter including nausea, lethargy, and poor coordination [20,21,22,23]. A very unpleasant odor is present at 0.3 ppm [24], and this is a valuable warning sign. The closest residents were 1.34 km away from the leak area. The Carris and ALOHA atmospheric diffusion models of the first leak (17 May) yielded AEGL-1, -2, and -3 radii of 2.8, 0.97, and 0.33 km, and resident exposure levels were low. Residents were exposed to concentrations below AEGL-1, and the second leak was less serious than the first. At or below the AEGL-1 level, irritation would not be expected [25]. The maximum styrene level was less than AEGL-2, and no AEGL-3 exposure occurred in residential areas. Although significant health effects were thus not expected, many residents complained that their symptoms persisted for up to 6 months after exposure. As a result of identifying the health effects of the SM leakage accident on residents, it is certain that all 1201 people who participated in the health effect survey smelled the smell caused by exposure. In addition, it was determined that it is certain that 1197 people, excluding 4 of them, had mild or moderate irritation symptoms due to the smell.

Acute chemical exposures (such as spills) commonly induce irritation, long-term chronic disease, anxiety, depression, and somatization [26]. Some studies have suggested that preexisting psychiatric factors may play roles in symptom persistence after exposure to major toxins [26]. Chemical spills not only induce physical problems but also mental conditions including posttraumatic embitterment disorder (PTED), major depression, and anxiety. It is difficult to determine whether leaks cause the mental health problems or exacerbate poor existing mental health [27]. Predisaster mental health assessments are not usually available, because it is impossible to predict when a disaster will occur. However, those with prior mental health problems may be more sensitive to new or worsening physical symptoms after an accident. Prior psychological problems are strong predictors of additional difficulties after a disaster [28]. We found that prior physical and mental health symptoms were the strongest predictors of symptom persistence after exposure to SM. Poor mental health status predicts postoperative pain persistence [29]. Depression increases the likelihood of poor outcomes in individuals with nonfatal conditions. Interactions among numerous risk factors may explain problems such as prior major depressive disorder [30]. Our findings are consistent with these results.

Individuals affected by leakages may seek compensation for mental and physical harm, and systemic symptoms are more frequent in those seeking compensation [31,32]. PTED is a reactive disorder induced by negative life events, and associated mental health problems may be highly persistent [33]. Seosan has an industrial complex with many factories (including chemical factories). Residents are chronically exposed to low concentrations of various substances that may affect the central nervous and respiratory systems, as well as the eyes and skin. We suggest that living in such an area is associated with perceived risks to health and safety, which may cause anxiety, insomnia, and depression [34,35]. The chemical disaster at the Cantara Loop of the Sacramento River triggered severe psychological, psychosocial, psychophysiological, and physical reactions in exposed compared to unexposed residents [36]. At the time of investigation (6 months after the leak), affected residents experienced more depression, anxiety, and physical symptoms. The media imprinted and consolidated memories of the disaster: long-term negative media coverage may induce posttraumatic stress-related symptoms. As media attention increased, so too did medically unexplained somatic symptoms [37]. However, we found no literature on the impact of news media on PTED.

This study had both strengths and limitations. As a strength, all residential areas within 10 km of the accident were included, and an exposure assessment was performed. We also recorded symptoms present before and within 24 h of the accident and at the time of the survey (average of 6 months later). Thus, we determined whether existing symptoms were exacerbated or new symptoms developed. Although we assessed psychiatric problems, we did not use a recognized scale (such as the Minnesota Multiphasic Personality Inventory); rather, we used the ACE toolkit, which is designed to evaluate accidents. However, reliability analysis was not performed on the main investigation items, such as the subject’s symptoms. The interview survey used in this study may have introduced several biases. A bias is likely to arise from persons failing to ascertain the onset time for exposure or symptoms [38]. Respondents can optionally hide personal symptoms or medical history information [39]. Some questions may be structured to cause respondents to select incorrect answers [40]. This may be the effect of memory decay bias that occurred after more than 5 months of the health effect survey. In addition, no biomarker or clinical definition of styrene toxicity exists, making it difficult to distinguish the effects of styrene from other effects. Finally, we only explored symptoms, which are somewhat subjective and may not fully reflect SM exposure.

## 5. Conclusions

In conclusion, prior mental health-related symptoms are risk factors for symptom persistence after chemical disaster. Our study indicates that after a disaster, management of individuals with preaccident mental symptoms or disease is needed. Even if it is not a large-scale disaster, such mental health care is important in chemical accidents that cause a lot of health concerns among residents.

## Figures and Tables

**Figure 1 healthcare-11-01004-f001:**
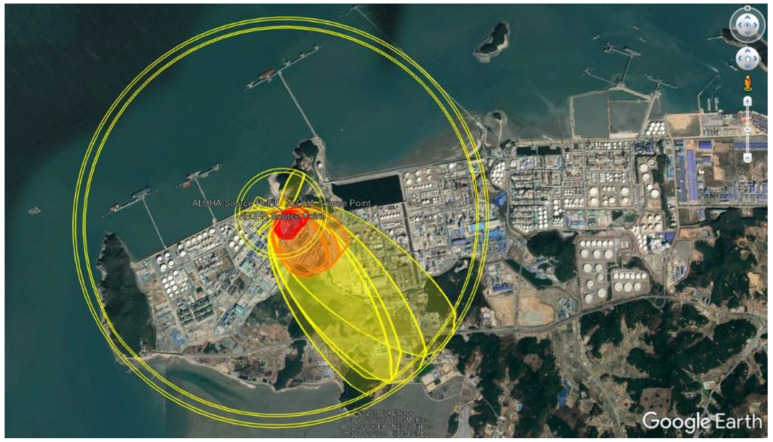
Area affected by the styrene monomer leakage.

**Figure 2 healthcare-11-01004-f002:**
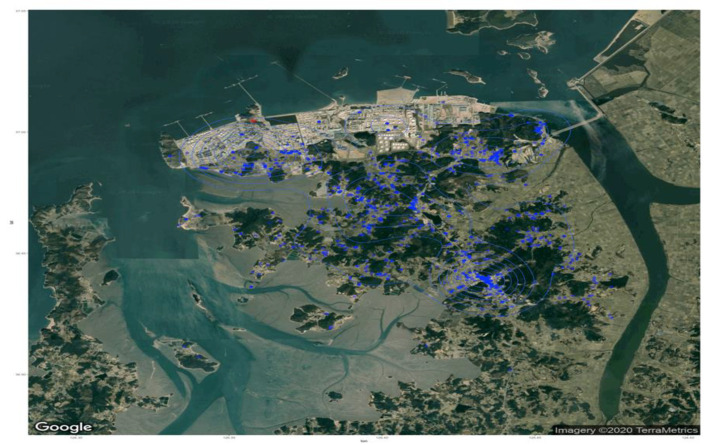
Geocoded locations of study participants at the time of exposure (blue spot). The spill area is indicated by the red spot.

**Table 1 healthcare-11-01004-t001:** Characteristics of study subjects.

		Men		Women		Unknown		Total	
		Number (Person)	Proportion (%)	Number (Person)	Proportion (%)	Number (Person)	Proportion (%)	Number (Person)	Proportion (%)
Age	<29	8	1.6	8	1.2	0	0	16	1.3
	30–39	23	4.4	15	2.4	0	0	38	3.2
	40–49	48	9.3	43	6.8	0	0	91	7.6
	50–59	124	24	131	20.7	0	0	255	21.2
	60–69	137	26.5	182	28.8	2	3.8	321	26.7
	≥70	177	34.2	253	40.3	0	0	430	35.8
	Unknown	0	0	0	0	50	96.2	50	4.2
Job	No	246	47.6	419	66.3 *	36	69.2	701	58.4
	Yes (No chemical exposure)	228	44.1	177	28	12	23.1	417	34.7
	Yes (Chemical exposure)	43	8.3	36	5.7	4	7.7	83	6.9
Distance from accident site	<3 km	149	28.8	88	13.9	6	11.5	243	20.2
	3–4.99 km	64	12.4	69	10.9	2	3.8	135	11.2
	5–9.99 km	224	43.3	329	52.1	29	55.8	582	48.5
	≥10 km	80	15.5	146	23.1	15	28.8	241	20.1
Location at the time of the accident	In door	114	22.1	193	30.5	17	32.7	324	27
	Out door	403	77.9 *	439	69.5	35	67.3	877	73
Odor intensity	Low	9	1.7	11	1.7	3	5.8	23	1.9
	Mild	23	4.4	21	3.3	3	5.8	47	3.9
	Moderate	69	13.3	66	10.4	6	11.5	141	11.7
	Severe	416	80.5	534	84.5	40	76.9	990	82.4
Evacuation status	No	310	60	415	65.7	37	71.2	762	63.4
	Yes	207	40	217	34.3	15	28.8	439	36.6

* *p* < 0.05 by chi-squared test.

**Table 2 healthcare-11-01004-t002:** Symptom prevalence by organ and time point.

Symptoms		Experienced Symptom within 24 h of the Incident		Experienced Symptom before the Incident		Symptom Worse after the Incident		Still Experiencing Symptom	
		Prevalence	%	Prevalence	%	Prevalence	%	Prevalence	%
Nervous system	Headache	1032	85.9	37	3.1	181	15.1	419	34.9
	Dizziness	1012	84.3	40	3.3	179	14.9	413	34.4
	Seizures	8	0.7	1	0.1	4	0.3	8	0.7
	Numbness pins and needles	429	35.7	41	3.4	101	8.4	244	20.3
	Difficulty concentrating	548	45.6	9	0.7	61	5.1	148	12.3
	Confusion	344	28.6	2	0.2	15	1.2	62	5.2
	Loss of balance	335	27.9	11	0.9	23	1.9	71	5.9
	Subtotal	1141	95.0	102	8.5	271	22.6	580	48.3
Respiratory	Breathing slow	110	9.2	4	0.3	5	0.4	25	2.1
	Breathing fast	462	38.5	10	0.8	22	1.8	67	5.6
	Difficulty breathing/feeling out-of-breath	678	56.5	20	1.7	75	6.3	195	16.2
	Coughing	837	69.7	24	2.0	115	9.6	308	25.6
	Increased congestion or phlegm	654	54.5	24	2.0	127	10.6	279	23.2
	Wheezing in chest	469	39.1	11	0.9	32	2.7	118	9.8
	Subtotal	1012	84.3	75	6.2	202	16.8	457	38.1
Musculoskeletal	Weakness of arms	485	40.4	24	2.0	76	6.3	190	15.8
	Weakness of legs	489	40.7	28	2.3	77	6.4	183	15.2
	Muscle twitching	238	19.8	9	0.7	28	2.3	90	7.5
	Tremors in arms or legs	358	29.8	12	1.0	55	4.6	127	10.6
	Generalized weakness	369	30.7	9	0.7	42	3.5	153	12.7
	Diffuse muscle aches and pains	382	31.8	22	1.8	90	7.5	188	15.7
	Subtotal	731	60.9	71	5.9	171	14.2	376	31.3
Cardiovascular	Slow heart rate/pulse	64	5.3	6	0.5	4	0.3	23	1.9
	Fast heart rate/pulse	392	32.6	5	0.4	14	1.2	51	4.2
	Chest tightness or pain/angina	521	43.4	14	1.2	87	7.2	205	17.1
	Blue or gray coloring of ends of fingers/toes or lips	66	5.5	4	0.3	7	0.6	23	1.9
	Subtotal	649	54.0	25	2.1	98	8.2	237	19.7
Gastrointestinal	Nausea	1036	86.3	16	1.3	121	10.1	278	23.1
	Vomiting	395	32.9	6	0.5	38	3.2	63	5.2
	Diarrhea	188	15.7	6	0.5	20	1.7	44	3.7
	Abdominal pain	316	26.3	7	0.6	21	1.7	65	5.4
	Subtotal	1099	91.5	64	5.1	205	17.1	409	34.1
Skin	Irritation, pain, or burning of skin	460	38.3	7	0.6	67	5.6	180	15
	Skin rash	239	19.9	5	0.4	42	3.5	107	8.9
	Skin blisters	99	8.2	5	0.4	20	1.7	37	3.1
	Sweating	449	37.4	19	1.6	31	2.6	59	4.9
	Cool or pale skin	231	19.2	4	0.3	8	0.7	32	2.7
	Skin discoloration	93	7.7	5	0.4	14	1.2	29	2.4
	Subtotal	705	58.7	40	3.3	101	8.4	264	22
Mental health	Anxiety	742	61.8	24	2.0	108	9.0	320	26.6
	Agitation/irritability	601	50.0	9	0.7	73	6.1	204	17
	Fatigue/tiredness	733	61.0	21	1.7	110	9.2	271	22.6
	Difficulty sleeping	712	59.3	27	2.2	156	13.0	336	28
	Feeling depressed	468	39.0	10	0.8	78	6.5	206	17.2
	Hallucinations	64	5.3	1	0.1	4	0.3	16	1.3
	Subtotal	923	76.9	70	5.8	216	18.0	508	42.3
Eye	Irritation/pain/burning	851	70.9	43	3.6	143	11.9	344	28.6
	Increased tearing	749	62.4	37	3.1	115	9.6	281	23.4
	Blurred vision	674	56.1	46	3.8	155	12.9	318	26.5
	Subtotal	984	81.9	89	7.4	229	19.1	476	39.6
Total		1197	99.7	355	29.6	536	44.6	853	71

**Table 3 healthcare-11-01004-t003:** Average number of symptoms by organ in subjects with mental health symptoms before the accident.

Organ Systems	Investigation Period	Mental Health Symptoms Prior to Accident		
		No	Yes	*p*-Value
Nerve system	Within 24-h of the accident	3.02	3.37	0.002
	Six months after accident	0.99	1.71	0.000
Musculoskeletal	Within 24-hof the accident	1.75	2.66	0.000
	Six months after accident	0.60	1.48	0.000
Respiratory	Within 24-hof the accident	2.75	3.51	0.000
	Six months after accident	0.79	1.25	0.000
Cardiovascular	Within 24-h of the accident	0.80	1.13	0.000
	Six months after accident	0.22	0.37	0.000
Gastrointestinal	Within 24-h of the accident	2.37	3.13	0.000
	Six months after accident	0.49	0.93	0.000
Skin	Within 24-h of the accident	1.23	1.63	0.000
	Six months after accident	0.33	0.53	0.001
Mental health	Within 24-h of the accident	2.56	3.59	0.000
	Six months after accident	0.83	2.32	0.000
Genitourinary	Within 24-h of the accident	0.24	0.55	0.000
	Six months after accident	0.11	0.45	0.000
Eye	Within 24-h of the accident	1.83	2.13	0.000
	Six months after accident	0.70	1.12	0.000
Nose	Within 24-h of the accident	1.10	1.30	0.000
	Six months after accident	0.23	0.34	0.000
Total	Within 24-h of the accident	18.02	23.46	0.000
	Six months after accident	5.39	10.63	0.000

**Table 4 healthcare-11-01004-t004:** Preexisting mental health symptoms affecting symptom persistence at the time of investigation.

Organ Systems	Adjusted for Preexisting Physical Symptoms *			Unadjusted for Preexisting Physical Symptoms *		
	Odds Ratio 95% C.I.	Nagelkerke R^2^	Hosmer and Lemeshow Test	Odds Ratio 95% C.I.	Nagelkerke R^2^	Hosmer and Lemeshow Test
Nervous system	1.54 (1.08–2.20)	0.180	0.113	3.17 (2.33–4.33)	0.226	0.245
Musculoskeletal	1.92 (1.32–2.79)	0.206	0.172	4.95 (3.65–6.71)	0.264	0.146
Respiratory	1.04 (0.71–1.51)	0.191	0.376	2.70 (2.02–3.62)	0.143	0.781
Cardiovascular	0.78 (0.49–1.22)	0.150	0.408	2.19 (1.58–3.03)	0.133	0.269
Gastrointestinal	1.45 (1.02–2.05)	0.153	0.454	3.01 (2.25–4.04)	0.134	0.130
Skin	0.82 (0.54–1.23)	0.156	0.485	1.84 (1.34–2.54)	0.126	0.519
Urogenital	1.21 (0.70–2.08)	0.237	0.419	4.04 (2.80–5.82)	0.145	0.178
Eye	0.98 (0.68–1.41)	0.197	0.707	2.26 (1.69–3.02)	0.148	0.352
Nose	0.95 (0.63–1.43)	0.149	0.798	1.85 (1.33–2.58)	0.134	0.505
Ear	0.49 (0.27–0.90)	0.120	0.221	1.22 (0.78–1.89)	0.118	0.694
Total	5.47 (2.57–11.65)	0.241	0.125	15.29 (7.45–31.36)	0.193	0.114

* Adjusted for sex, location at the time of the accident, odor intensity, distance from accident site, evacuation status, and age.

## Data Availability

Not applicable.

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
