# Peer review of "The Effect of Prior Mental Health on Persistent Physical Symptoms after Exposure to a Chemical Disaster"

_healthcare, 2023, doi:10.3390/healthcare11071004_

Round 1

Reviewer 1 Report

See attached file for comments.

Author Response

We are grateful for your scrupulous attention to detail and the insightful comments. We agree to your comments that current version of our manuscript should be revised according to reviewer’s comments and to add more clarity to our findings. The changes are highlighted below. We would like to express our sincere gratitude to you. Thank you.

Reviewer 2 Report

Thank you for the opportunity to review this manuscript. It is an interesting topic. 

On the one hand, the article seems to be very concise (it may seem too short), and on the other hand, the essential elements of a scientific article have been covered: introduction, purpose of the study, material, methods and tools used in the study and results. In my opinion, the discussion could be expanded, but the authors themselves admit that there is little research of this type. They also admit that the study has its limitations. 

What I noticed and in my opinion should be changed is the title. I think it should not be used to claim that "something" is "something" - in this case "previous mental health symptoms" are "risk factors". I propose a change.

Author Response

(The authors gave the same response as above.)

Reviewer 3 Report

The manuscript entitled "Prior mental health symptoms are risk factors for persistent physical symptoms after exposure to a chemical disaster" describes a study focusing on the psychological predictors of functioning after chemical disaster. The data and design of the study are unique and could help in understanding reaxtions to chemical disasters. However, the manuscript its self needs some modifications:

#1. The introduction should be not only the description o the particular disaster studied, but also should include information about previous findings on the reactions toward similar disasters. These data is discussed later.

#2. The procedure of the study should be described in details. Precisely, when and how were the participants invited and contacted. When the mental symptoms examined (prior to the disaster)?

#3. The general statistics about regression model are lacking (pseudo R2; statistics for models).

#4. Some corections for multiple comparisons should be included, e.g. Bonferroni correcteion.

#5. The division of the participants present in Table 3 should be described. How the participants were divided? Using the cut-off of at least one symptom? Why?

#6. Did the Authors analyse the reliability of the measure?

Author Response

(The authors gave the same response as above.)

Round 2

Reviewer 3 Report

Thank to the Authors for their revision. They addressed all my previous suggestions. In my opinion, they can also include data on internal reliability of the measure to the manuscript and point that cardiovascular, urogenital and nose symptoms are a little below the typical treshold for internal consistency (namely .60). In general, I think that the manuscript is an important contribution regarding the specifity of the disaster which was studied.